

# Identification of key genes and biological pathways in Chinese lung cancer population using bioinformatics analysis

Ping Liu[1,*], Hui Li[1,*], Chunfeng Liao[2], Yuling Tang[1], Mengzhen Li[3], Zhouyu Wang[3], Qi Wu[4] and Yun Zhou[5]

[1] Department of Respiratory Medicine, The First Hospital of Changsha, Changsha, China
[2] Department of Cardiology, The First Hospital of Changsha, Changsha, China
[3] MyGene Diagnostics Co., Ltd., Guangzhou, China
[4] Department of Emergency, The First Hospital of Changsha, Changsha, China
[5] Department of Spinal Surgery, The First Hospital of Changsha, Changsha, China
[*] These authors contributed equally to this work.

Corresponding authors
Qi Wu, 392554098@qq.com
Yun Zhou, zhouyun83@sina.com

## ABSTRACT

**Background**. Identification of accurate prognostic biomarkers is still particularly urgent for improving the poor survival of lung cancer patients. In this study, we aimed to identity the potential biomarkers in Chinese lung cancer population via bioinformatics analysis.

**Methods**. In this study, the differentially expressed genes (DEGs) in lung cancer were identified using six datasets from Gene Expression Omnibus (GEO) database. Subsequently, enrichment analysis was conducted to evaluate the underlying molecular mechanisms involved in progression of lung cancer. Protein-protein interaction (PPI) and CytoHubba analysis were performed to determine the hub genes. The GEPIA, Human Protein Atlas (HPA), Kaplan-Meier plotter, and TIMER databases were used to explore the hub genes. The receiver operating characteristic (ROC) analysis was performed to evaluate the diagnostic value of hub genes. Reverse transcription quantitative PCR (qRT-PCR) was used to validate the expression levels of hub genes in 10 pairs of lung cancer paired tissues.

**Results**. A total of 499 overlapping DEGs (160 upregulated and 339 downregulated genes) were identified in the microarray datasets. DEGs were mainly associated with pathways in cancer, focal adhesion, and protein digestion and absorption. There were nine hub genes (CDKN3, MKI67, CEP55, SPAG5, AURKA, TOP2A, UBE2C, CHEK1 and BIRC5) identified by PPI and module analysis. In GEPIA database, the expression levels of these genes in lung cancer tissues were significantly upregulated compared with normal lung tissues. The results of prognostic analysis showed that relatively higher expression of hub genes was associated with poor prognosis of lung cancer. In HPA database, most hub genes were highly expressed in lung cancer tissues. The hub genes have good diagnostic efficiency in lung cancer and normal tissues. The expression of any hub gene was associated with the infiltration of at least two immune cells. qRT-PCR confirmed that the expression level of CDKN3, MKI67, CEP55, SPAG5, AURKA, TOP2A were highly expressed in lung cancer tissues.

**Conclusions**. The hub genes and functional pathways identified in this study may contribute to understand the molecular mechanisms of lung cancer. Our findings may provide new therapeutic targets for lung cancer patients.

## INTRODUCTION

Lung cancer has become the most common type of cancer in the world, leading to the largest number of cancer-related deaths (*Siegel, Miller & Jemal, 2019*). More than 80% of lung cancer are non-small cell lung cancer (NSCLC), mainly lung adenocarcinoma (LUAD) and lung squamous cell carcinoma (LUSC) (*Travis, 2011*). LUAD accounts for over 70% of NSCLC (*Hirsch et al., 2016*). Smoking is the most important risk factor for lung cancer, many other lifestyle and occupational factors also have a significant impact (*Parida, Siddharth & Sharma, 2021*). Changes in risk factors for cancer, especially diet, obesity, diabetes, and air pollution, continue to fuel the trend of cancer transformation in China (*Sun et al., 2020*). Although there are various treatment methods for lung cancer, including surgery, chemotherapy, radiotherapy, targeted therapy, immunotherapy, and palliative treatment, the 5-year survival rate of lung cancer in recent decades is still very low (*Wang, Chen & Liu, 2020*). Therefore, the identification of accurate prognostic biomarkers and novel therapeutic targets is still particularly urgent for improving the poor survival of NSCLC patients.

In recent years, with the development of microarray and high-throughput sequencing technologies, a large number of open data resources, such as the Cancer Genome Atlas Database (TCGA) and Gene Expression Comprehensive Database (GEO), have generated a large amount of gene data (*Yu & Tian, 2020*). Bioinformatics can effectively screen and mine microarray data, thereby revealing potential oncogenes at the molecular level (*Gu et al., 2018*; *Liu et al., 2019*). Bioinformatics has been widely used to find molecular markers and signaling pathways related to the occurrence and development of lung cancer (*Jiao et al., 2020*; *Li, Qi & Li, 2020*; *Li et al., 2020*). Possibly driven by the community and environmental factors, the observed differences in disease incidence suggest the importance of residential location in risk assessment of lung cancer (*Zhu et al., 2020*). At present, most of the GEO datasets used for lung cancer research are from different countries (*Song, Tang & Li, 2021*).

In this study, the differentially expressed genes (DEGs) in lung cancer were identified using 6 GEO datasets from Chinese population. Then, functional enrichment analysis was conducted to evaluate the underlying molecular mechanisms involved in progression of lung cancer. Subsequently, we conducted protein-protein interaction (PPI) and CytoHubba analysis to identify the potential hub genes associated with lung cancer. The GEPIA, Human Protein Atlas (HPA), Kaplan–Meier plotter, and TIMER databases were used to explore the hub genes. ROC analysis was performed to evaluate the diagnostic value of hub genes. Finally, Reverse transcription quantitative PCR (qRT-PCR) was used to validate the expression levels of hub genes in 10 pairs of lung cancer paired tissues. Our research will provide some useful biomarkers for the diagnosis and prognosis of lung cancer.

**Table 1  Characteristics of the six GEO datasets.**

| GEO ID | Platform | Normal samples (*n*) | Tumor samples (*n*) |
|--------|----------|----------------------|---------------------|
| GSE136043 | GPL13497 | 5 | 5 |
| GSE130779 | GPL20115 | 8 | 8 |
| GSE118370 | GPL570 | 6 | 6 |
| GSE85841 | GPL20115 | 8 | 8 |
| GSE85716 | GPL19612 | 6 | 6 |
| GSE89039 | GPL17077 | 8 | 8 |

Notes.

GEO, Gene Expression Omnibus.

# MATERIALS AND METHODS

## The information of GEO datasets

Six datasets, including GSE136043, GSE130779, GSE118370, GSE85841, GSE85716, and GSE89039 were selected from GEO database (https://www.ncbi.nlm.nih.gov/geo/). The inclusion criteria for the above datasets were set as follows: (1) The samples of the datasets were all from China; (2) the datasets included human lung cancer tissues and normal tissues; (3) the number of samples in each dataset was more than 10. The GSE136043 dataset included five LUAD samples and five normal samples. The GSE130779 dataset included eight LUAD samples and eight normal samples. The GSE118370 dataset included six LUAD samples and six normal samples. The GSE85841 dataset included eight LUAD samples and eight normal samples. The GSE85716 dataset included six LUAD samples and six normal samples. The GSE89039 dataset included eight LUAD samples and eight normal samples. Six datasets included a total of 41 LUAD tissues and 41 normal lung tissues (Table 1).

## Identification of DEGs

R packages (GEOquery and dplyr) were performed to match the expression matrix to the probe (*Li, Qi & Li, 2020*; *Li et al., 2020*). The DEGs in each microarray were filtrated by the limma package. RobustRankAggreg (RRA) was used to integrate the DEGs identified from six datasets (*Kolde et al., 2012*). The RRA algorithm can handle a variable number of genes identified from different microarray platforms. Next, | log 2 FC | > 1.0 and adjusted *P*-value < 0.05 were used to filtrate the DEGs.

## GO and KEGG enrichment analysis

The DAVID database (v6.8, https://david.ncifcrf.gov/) was used to perform the Gene Ontology (GO) and Kyoto Encyclopedia of Genes and Genomes (KEGG) enrichment analysis (*Huang, Sherman & Lempicki, 2009*). The results of GO annotation contain three parts, including biological process (BP), cell component (CC), and molecular function (MF). The top 15 GO terms were listed according to *P*-value. The results were considered statistically significant if *P* < 0.05. The KEGG pathways were visualized by ggplot2 package (*P* < 0.05) (*Tang et al., 2020*).

### PPI network construction and hub genes identification

The protein-protein interactions of the overlapping DEGs were obtained *via* the STRING database (https://string-db.org/) (*Szklarczyk et al., 2019*). The combined score of medium confidence > 0.4 was used as the cut-off value in the STRING database. Subsequently, a clear illustration of the PPI was demonstrated using Cytoscape software (v3.8.0) with CytoHubba, which is a plug-in that uses the degree algorithm to screen the hub genes (*Ma et al., 2020*). The degree, edge percolated component (EPC), maximal clique centrality algorithm (MCC), and maximum neighborhood component (MNC) algorithms in CytoHubba were used to select the hub genes (*Ma et al., 2021*). The top 20 nodes with the degree, EPC, MCC and MNC were selected, and we take the intersection of the four algorithm as the hub genes. The Cytoscape plug-in Molecular Complex Detection (MCODE) (degree cutoff = 2, Node Score Cutoff = 0.2, and K −Core = 2) was used to capture the hub network modules (*Dai et al., 2020*).

### Validation of mRNA expression levels of hub genes

The GEPIA (http://gepia.cancer-pku.cn/index.html) is an online database that consists of 9,736 tumors and 8,587 normal samples from TCGA and GTEx data (*Mou et al., 2021*). The mRNA expression levels of hub genes were validated by GEPIA database.

### Validation of hub genes via Kaplan Meier plotter database

The identification of overall survival (OS) rates of hub genes in LUAD was performed using the Kaplan Meier plotter database (https://kmplot.com/analysis), an online tool used to assess the effect of 54 k genes on survival across 21 cancer types (*Liu et al., 2020*). The Kaplan–Meier plotter is an online deposit of the survival analysis data of EGA, TCGA, and GEO (Affymetrix microarrays only) databases (*Yang et al., 2020*). A log-rank $P < 0.05$ was considered to be statistically significant.

### Validation of protein expression levels of hub genes in HPA database

The protein expression levels of hub genes in lung cancer tissues and normal tissues were validated using immunohistochemistry (IHC) results from the HPA database (https://www.proteinatlas.org/) (*Li, Qi & Li, 2020*; *Li et al., 2020*).

### Validation of hub genes by ROC analysis

Then, we performed a receiver operating characteristic (ROC) analysis using TCGA database to evaluate the diagnostic value of hub genes (*Jia et al., 2021*). Usually, the AUC value > 0.85 showed a good diagnostic value for lung cancer.

### Immune infiltrates analysis of hub genes

Tumor immune estimation resource (TIMER) (https://cistrome.shinyapps.io/timer/) is a comprehensive website for systematic analysis of tumor infiltrating immune cells of 32 different cancers in TCGA database (*Yang et al., 2021*). In this study, TIMER database was used to estimate the associations between hub genes expression and immune cell populations (B cells, CD8 + T cells, CD4 + T cells, macrophages, neutrophils and dendritic cells) in LUAD.

## Lung cancer tissues

Tumor and adjacent normal tissues were obtained from 10 lung cancer patients in the The First Hospital of Changsha between September 2021 and October 2021. These patients had no other major illnesses. A total of 20 frozen tissue specimens contained 10 tumor tissues and 10 matched adjacent non-tumor tissues were obtained. The detailed clinical information of the patients is shown in Table S1. All tissues were preserved and stored at −80 °C. The study was approved by the Ethics Committee of The First Hospital of Changsha and informed consent was obtained from all patients.

## Quantitative real-time reverse transcription PCR (qRT-PCR)

Total RNA from lung cancer patients' tissues was isolated by TRIzol reagent (Invitrogen, Carlsbad, CA, USA). Real-time PCR was further performed with SYBR Green Master Mix (Takara, Japan) according to the manufacturer's protocols. The primers used in this study are provided in Table S2. GAPDH were used as an internal control and $2^{-\Delta\Delta Ct}$ method was applied to evaluate gene expression levels. Differences in relative expression levels were analysed through t test or M ann-Whitney U test (SPSS, V 22.0). The results were considered statistically significant if $P < 0.05$.

# RESULTS

## Identification of DEGs among six GEO datasets

The LUAD chip expression datasets GSE136043, GSE130779, GSE118370, and GSE85841, GSE85716, and GSE89039 were normalized (Fig. S1). The GSE136043 contained 1,206 DEGs, including 541 upregulated genes and 665 downregulated genes. The GSE130779 dataset contained 2,964 DEGs, including 1,231 upregulated genes and 1,733 downregulated genes. The GSE118370 dataset contained 789 DEGs, including 211 upregulated genes and 578 downregulated genes. The GSE85841 dataset contained 3,058 DEGs, including 1,270 upregulated genes and 1,788 downregulated genes. The GSE85716 dataset contained 1,196 DEGs, including 367 upregulated genes and 829 downregulated genes. The GSE89039 dataset contained 2,966 DEGs, including 1,129 upregulated genes and 1,837 downregulated genes. The volcano plots of DEGs in the six datasets are shown in Fig. 1. A total of 499 DEGs were obtained through the RRA method, including 160 upregulated genes (Table S3) and 339 downregulated genes (Table S4). The top 20 up- and down-regulated genes after the integrated analysis are shown in Fig. 2.

## Functional enrichment analysis of overlapping DEGs

The 499 overlapping DEGs were subjected to the GO and KEGG enrichment analysis. The top 15 enriched GO terms from biological process, cellular component, and molecular function are shown in Fig. 3A. In the biological process, the DEGs were mainly enriched in angiogenesis, collagen catabolic process, and cell adhesion. In the cellular component, the DEGs were mainly enriched in extracellular region, proteinaceous extracellular matrix, and extracellular space. In the molecular function, the DEGs were mainly enriched in heparin binding, calcium ion binding, and metalloendopeptidase activity. For the KEGG pathways analysis, the DEGs were mainly enriched in pathways in cancer, focal adhesion, and protein digestion and absorption (Fig. 3B).

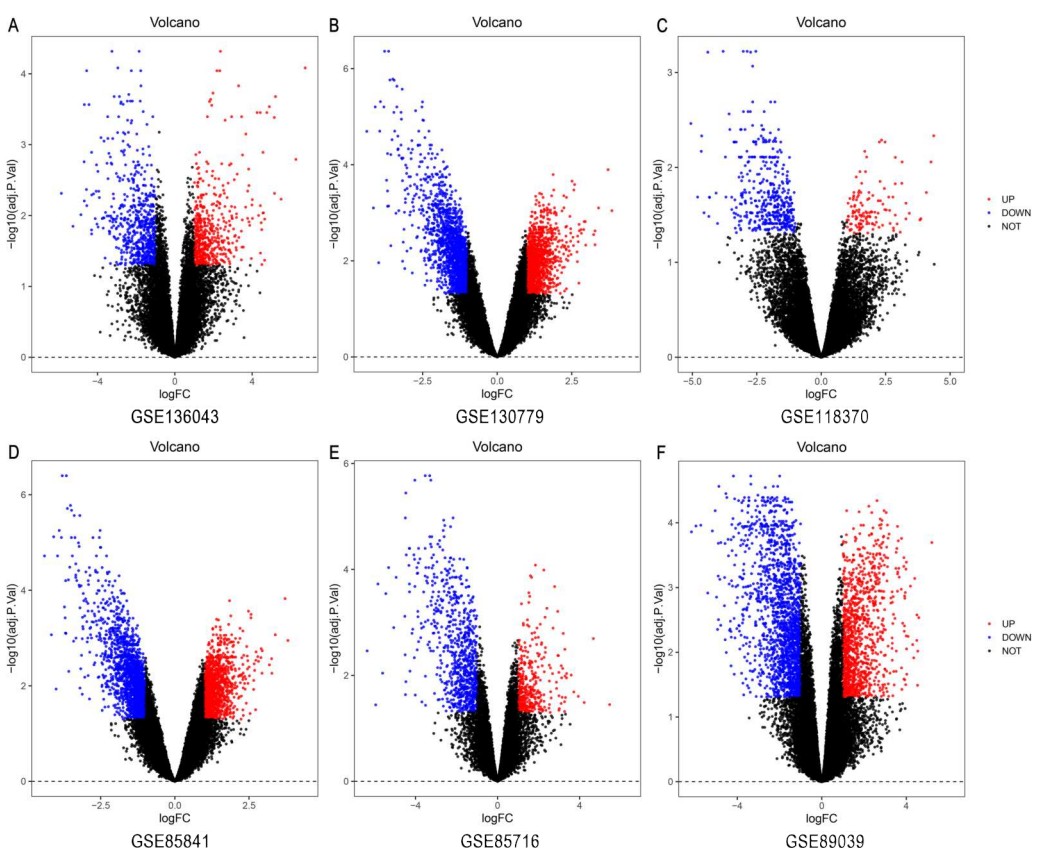

**Figure 1 The volcano plots of DEGs in six datasets.** The DEGs in (A) GSE136043 (B) GSE130779, (C) GSE118370, and (D) GSE85841, (E) GSE85716 and (F) GSE89039 datasets. The red dots represent upregulated genes according to an adjusted $P < 0.05$ and | log fold change | $> 1$; the blue dots represent downregulated genes according to an adjusted $P < 0.05$ and | log fold change | $> 1$; the black dots represent genes with no significant difference in expression. DEG, differentially expressed genes.

## PPI network construction and hub genes identification

The PPI network included 423 nodes and 1,331 edges (Fig. 4A). We interacted the results of four algorithms to improve the reliability of the hub genes (Table 2). A total of nine genes (*CDKN3, MKI67, CEP55, SPAG5, AURKA, TOP2A, UBE2C, CHEK1* and *BIRC5*) were considered hub genes. The top three modules from MCODE were selected for future analysis. Module 1 included 21 upregulated genes, including *TOP2A, INCENP, UBE2C, BIRC5, AURKA, TRIP13, CENPM, MND1, RAD54L, STIL, CHEK1, KIF14, GTSE1, CDC25C, MKI67, CEP55, MELK, CDKN3, SPAG5, CENPF, KIF20A* (Fig. 4B). Module 2 included 14 upregulated genes and five downregulated genes (Fig. 4C). Module 3 included one upregulated gene and five downregulated genes (Fig. 4D). The functional enrichment analysis of genes in module 1 were conducted by DAVID. These genes were significantly enriched in cell division, midbody and ATP binding (Table 3).

| GSE136043 | GSE130779 | GSE118370 | GSE85841 | GSE85716 | GSE89039 | Gene |
|---|---|---|---|---|---|---|
| 6.26 | 3.40 | 3.23 | 3.39 | 4.68 | 2.27 | SPP1 |
| 2.54 | 2.47 | 2.55 | 2.44 | 1.86 | 2.87 | HABP2 |
| 6.74 | 1.90 | 2.19 | 1.89 | 2.57 | 2.86 | CEACAM5 |
| 5.49 | 2.04 | 2.13 | 2.03 | 1.94 | 3.67 | COL10A1 |
| 3.30 | 2.44 | 0.95 | 2.42 | 3.32 | 3.83 | PITX1 |
| 1.95 | 3.72 | 1.98 | 3.72 | 1.80 | 3.92 | CST2 |
| 3.04 | 2.42 | 2.21 | 2.40 | 2.12 | 0.00 | SERPIND1 |
| 3.76 | 0.71 | 3.32 | 0.68 | 3.10 | 3.98 | TOX3 |
| 1.95 | 1.91 | 1.54 | 1.89 | 2.33 | 2.30 | STK32A |
| 4.58 | 1.80 | 2.61 | 1.76 | 0.46 | 3.23 | TMEM63C |
| 2.99 | 2.30 | 2.12 | 2.26 | 1.75 | 1.34 | CDH3 |
| -1.22 | 3.28 | 1.44 | 3.26 | 0.00 | 5.21 | GCNT3 |
| 3.96 | 0.68 | 2.74 | 0.66 | 2.87 | 4.30 | CXCL13 |
| 3.19 | 1.88 | 0.11 | 1.86 | 3.54 | 2.51 | EEF1A2 |
| 3.17 | 2.68 | 1.97 | 2.66 | 2.34 | 0.60 | LRRC31 |
| 1.99 | 3.28 | 2.65 | 3.27 | 3.10 | 1.10 | ABCC3 |
| 2.75 | 2.70 | 0.21 | 2.67 | 4.23 | 2.44 | TUBB3 |
| 2.54 | 2.75 | 1.93 | 2.74 | 1.82 | -0.16 | CXCL14 |
| 2.08 | 1.77 | 2.35 | 1.76 | 0.00 | 3.58 | CST1 |
| 3.16 | 2.32 | 0.00 | 2.31 | 5.47 | 0.94 | XAGE1A |
| -4.03 | -3.60 | -3.37 | -3.62 | -5.60 | -4.57 | CLDN18 |
| -3.82 | -2.38 | -3.03 | -2.39 | -4.49 | -4.41 | AGER |
| -4.07 | -2.59 | -3.28 | -2.61 | -3.86 | -3.36 | RTKN2 |
| -3.61 | -3.92 | -1.93 | -3.90 | -4.29 | -4.93 | ADAMTS8 |
| -3.59 | -3.68 | 0.00 | -3.68 | -5.06 | -4.30 | FIGF |
| -3.50 | -3.02 | -3.02 | -3.04 | -4.30 | -3.14 | GRIA1 |
| -3.33 | -2.27 | -4.55 | -2.30 | -4.84 | -3.05 | TMEM100 |
| -4.56 | -2.16 | -4.47 | -2.19 | -4.03 | -6.17 | SGCG |
| -3.06 | -2.14 | -3.23 | -2.16 | -3.52 | -3.06 | LYVE1 |
| -3.22 | -2.13 | -2.62 | -2.16 | -3.41 | -3.38 | STXBP6 |
| -2.39 | -3.18 | -2.28 | -3.20 | -5.45 | -4.18 | FAM107A |
| -2.36 | -2.93 | -2.33 | -2.95 | -4.47 | -3.14 | ADRB1 |
| -2.33 | -3.67 | -2.34 | -3.69 | -3.93 | -4.18 | TCF21 |
| -5.87 | -2.89 | 0.00 | -2.91 | -4.52 | -5.73 | ITLN2 |
| -3.18 | -2.01 | -2.35 | -2.04 | -3.19 | -3.11 | CDO1 |
| -2.20 | -3.19 | -2.50 | -3.20 | -4.03 | -2.90 | FMO2 |
| -3.46 | -2.63 | -2.26 | -2.64 | -2.09 | -3.99 | RBP2 |
| -2.14 | -2.16 | -2.48 | -2.18 | -3.85 | -3.48 | SOX7 |
| -2.94 | -3.78 | -1.49 | -3.79 | -3.78 | -4.34 | ADH1A |
| -3.15 | -2.14 | -1.95 | -2.18 | -4.17 | -3.65 | MYZAP |

**Figure 2   The top 20 up- and downregulated genes in integrated datasets.** The abscissa represents the GEO datasets, and the ordinate represents the gene name. The red represents log FC > 0; the pink represents log FC is slightly less than 0; the blue represents log FC < 0.

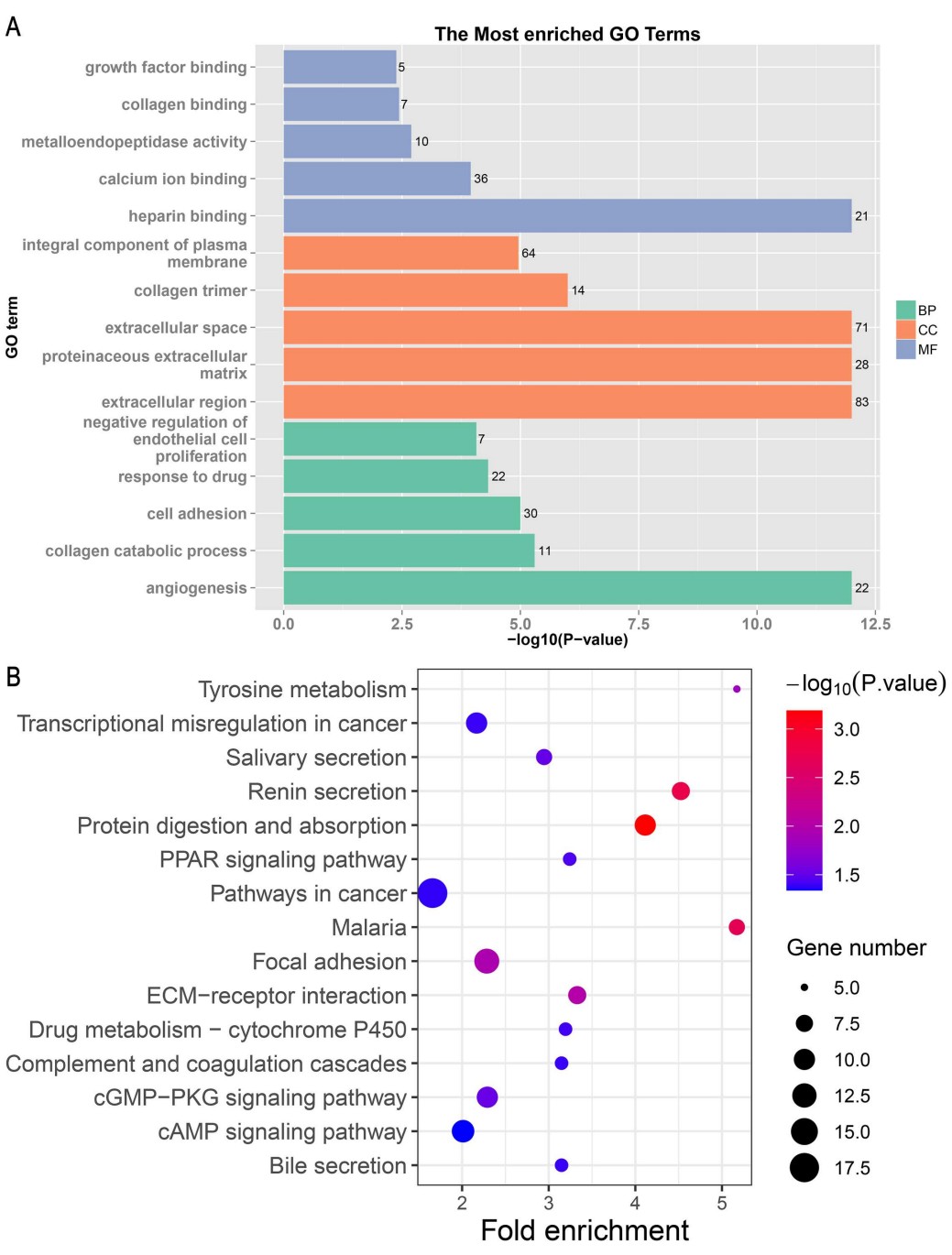

**Figure 3** **The results of enrichment analysis.** (A) The results of GO annotation analysis. (B) The KEGG pathway enrichment analysis of DEGs. GO, Gene ontology; KEGG, Kyoto Encyclopedia of Genes and Genomes; DEGs, differentially expressed genes.

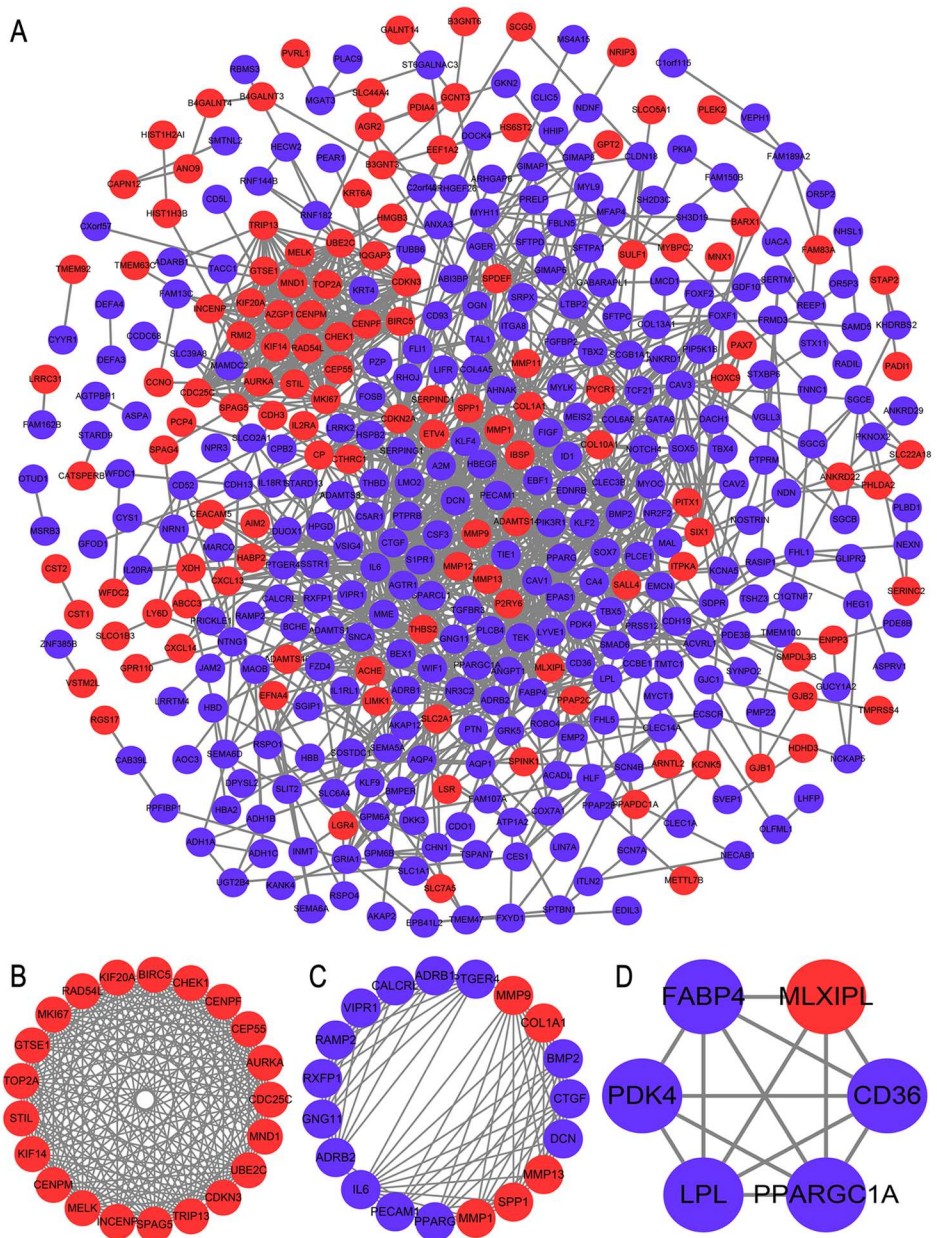

**Figure 4** **PPI network construction and module analysis.** (A) The PPI network of DEGs. The red circles represents the upregulated DEGs and blue circles represents the downregulated DEGs. (B) Module 1 from the PPI network. (C) Module 2 from the PPI network. (3) Module 3 from the PPI network. DEGs, differentially expressed genes; PPI, protein-protein interaction.

**Validation of hub genes by GEPIA and Kaplan Meier plotter database**

We validated mRNA expression levels of hub genes in the LUAD cohorts from GEPIA database. The LUAD cohort included 483 tumor tissues and 347 normal tissues. The mRNA expression levels of hub genes were higher in LUAD tissues than in normal lung tissues

**Table 2  The scored top 20 genes in Degree, EPC, MCC, and MNC algorithms.**

| Category | Rank methods in cytoHubba | | | |
|---|---|---|---|---|
| | Degree | EPC | MCC | MNC |
| 1 | IL6 | IL6 | TOP2A | IL6 |
| 2 | MMP9 | MMP9 | UBE2C | MMP9 |
| 3 | PECAM1 | PECAM1 | CEP55 | PECAM1 |
| 4 | COL1A1 | SPP1 | SPAG5 | COL1A1 |
| 5 | UBE2C | COL1A1 | CENPF | BMP2 |
| 6 | BMP2 | PPARG | KIF20A | PPARG |
| 7 | PPARG | CTGF | MELK | UBE2C |
| 8 | CAV1 | MKI67 | CHEK1 | TOP2A |
| 9 | CTGF | CHEK1 | AURKA | MKI67 |
| 10 | TOP2A | TOP2A | BIRC5 | CTGF |
| 11 | MKI67 | UBE2C | MKI67 | CHEK1 |
| 12 | AURKA | CDKN3 | CDKN3 | AURKA |
| 13 | CEP55 | BIRC5 | KIF14 | SPP1 |
| 14 | SPP1 | CEP55 | TRIP13 | CAV1 |
| 15 | BIRC5 | BMP2 | RAD54L | BIRC5 |
| 16 | SPAG5 | AURKA | CENPM | KIF20A |
| 17 | CDKN3 | MELK | CDC25C | SPAG5 |
| 18 | CHEK1 | CENPF | GTSE1 | CDKN3 |
| 19 | CENPF | SPAG5 | MND1 | MELK |
| 20 | KIF20A | CDKN2A | STIL | CEP55 |

**Notes.**

Degree, node connect degree; EPC, edge percolated component; MCC, maximal clique centrality; MNC, maximal neighborhood component.

(Fig. 5). Results from the Kaplan Meier plotter revealed that relatively higher expression of hub genes was associated with poor prognosis of LUAD patients (Fig. 6).

## Protein expression levels of hub genes in HPA database

The protein expression levels of hub genes were explored using the HPA database. As the immunohistochemical information of *CDKN3* and *CHEK1* were not existed in HPA, we have only displayed the results of *MKI67, CEP55, SPAG5, AURKA, TOP2A, UBE2C* and *BIRC5* (Fig. 7). The protein levels of *MKI67, AURKA* and *BIRC5* were not detected in normal lung tissues, while the levels of these genes were high in lung cancer tissues. No expression of *CEP55* was observed in normal lung tissues, while medium expression of *CEP55* was observed in tumor tissues. The protein levels of *TOP2A* and *UBE2C* were low in normal lung tissues, while the levels of these genes were high in lung cancer tissues. *SPAG5* was found to have medium expression in LUAD tissues, while low expression was observed in normal lung tissues. The results of HPA database showed that most hub genes may be highly expressed in lung cancer tissues.

**Table 3  The top 15 enriched GO terms of genes in module 1.**

| Category | Term | Count | *P* Value |
| --- | --- | --- | --- |
| BP | Cell division | 7 | 0.000002 |
| BP | Mitotic nuclear division | 6 | 0.000009 |
| BP | G2/M transition of mitotic cell cycle | 5 | 0.000019 |
| BP | Chromosome segregation | 4 | 0.000069 |
| BP | Protein localization to centrosome | 3 | 0.000204 |
| CC | Midbody | 8 | 0.000000 |
| CC | Nucleoplasm | 13 | 0.000006 |
| CC | Gcentriole | 5 | 0.000006 |
| CC | Spindle | 5 | 0.000008 |
| CC | Chromosome, centromeric region | 4 | 0.000032 |
| MF | ATP binding | 10 | 0.000013 |
| MF | Protein binding | 19 | 0.000075 |
| MF | Protein kinase binding | 4 | 0.008149 |
| MF | Protein C-terminus binding | 3 | 0.017525 |
| MF | Microtubule binding | 3 | 0.022512 |

**Notes.**
BP, biological process; CC, cell component; MF, molecular function.

## The diagnostic value of hub genes in LUAD and their relationships with tumor infiltrating immune cells

The ROC curve was used to evaluate the diagnostic value of hub genes. As shown in Fig. 8, the AUC values of *CDKN3, MKI67, CEP55, SPAG5, AURKA, TOP2A, UBE2C, CHEK1* and *BIRC5* in LUAD were 0.965, 0.965, 0.980, 0.986, 0.975, 0.986, 0.984, 0.978, 0.980, respectively. Thus, the hub genes have good diagnostic efficiency in LUAD and normal tissues. The results from TIMER database showed that the 9 hub genes were not associated with tumor purity (Fig. 9). However, the expression of these genes was negatively correlated with B cell infiltration ($P < 0.05$). Only the *TOP2A* expression was associated with CD8+ T cells. The expression of *CDKN3, CEP55, AURKA* and *UBE2C* was related to the infiltration of macrophages. The expression of *AURKA* and *BIRC5* was related to the infiltration of dendritic cells. Thus, the expression of any hub gene was associated with the infiltration of at least two immune cells.

## Validation of the hub genes via qRT-PCR

The qRT-PCR was performed to further validate the expression of hub genes. As shown in Fig. 10, the relative expression levels of *CDKN3, MKI67, CEP55, SPAG5, AURKA, TOP2A* were consistent with the results of bioinformatics analysis ($P < 0.05$), while the expression levels of *UBE2C, CHEK1* and *BIRC5* in tumor samples were not significantly different from adjacent normal samples.

## DISCUSSION

Lung cancer is still a common cause of health issues worldwide (*Ma et al., 2020*). So far, many lung cancer studies based on gene arrays have been conducted by different

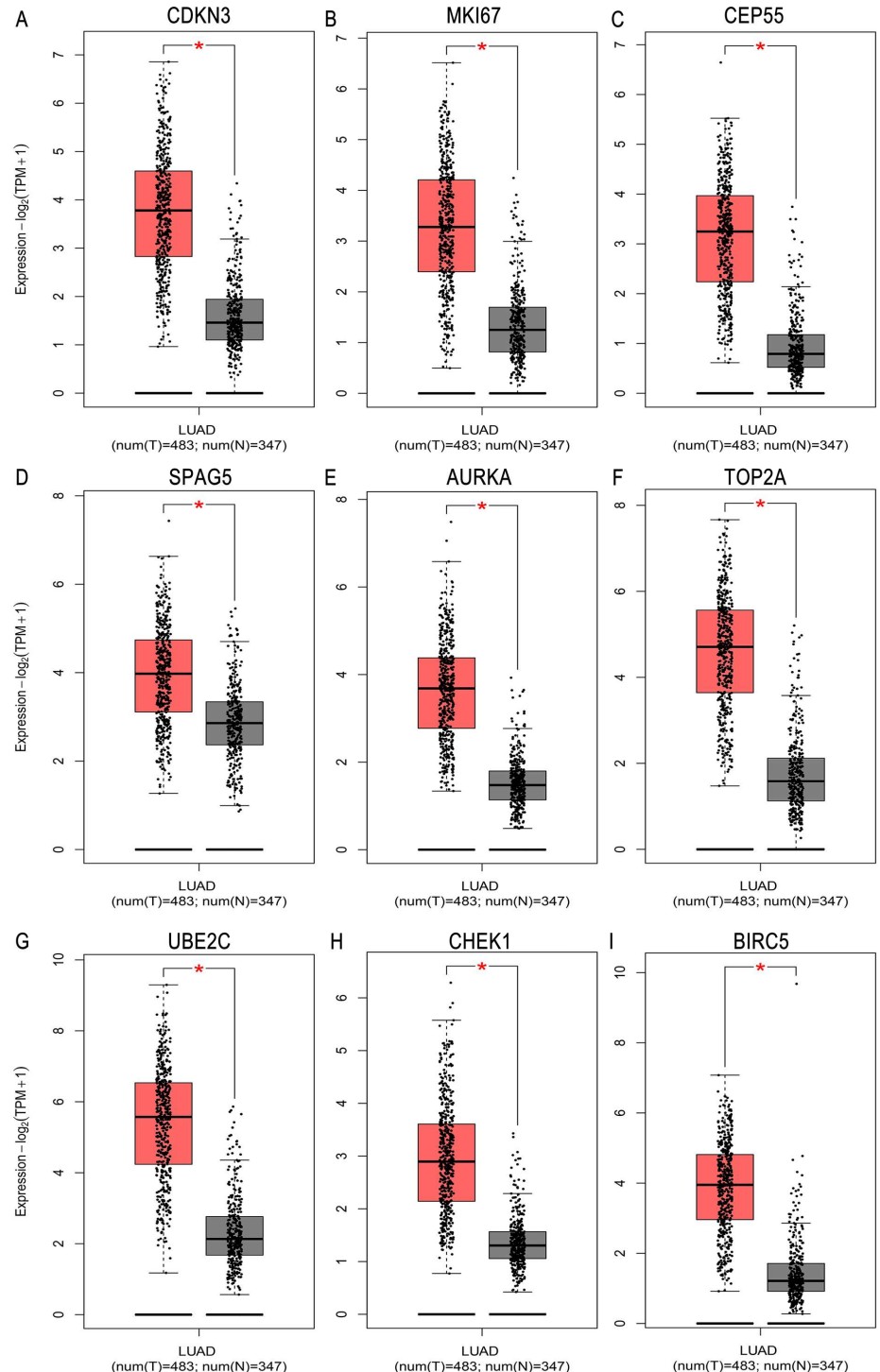

**Figure 5** **Validation of the mRNA expression of (A–I)** *CDKN3, MKI67, CEP55, SPAG5, AURKA, TOP2A, UBE2C, CHEK1,* **and** *BIRC5* **in LUAD tissues and normal tissues using GEPIA database.** LUAD, lung adenocarcinoma.

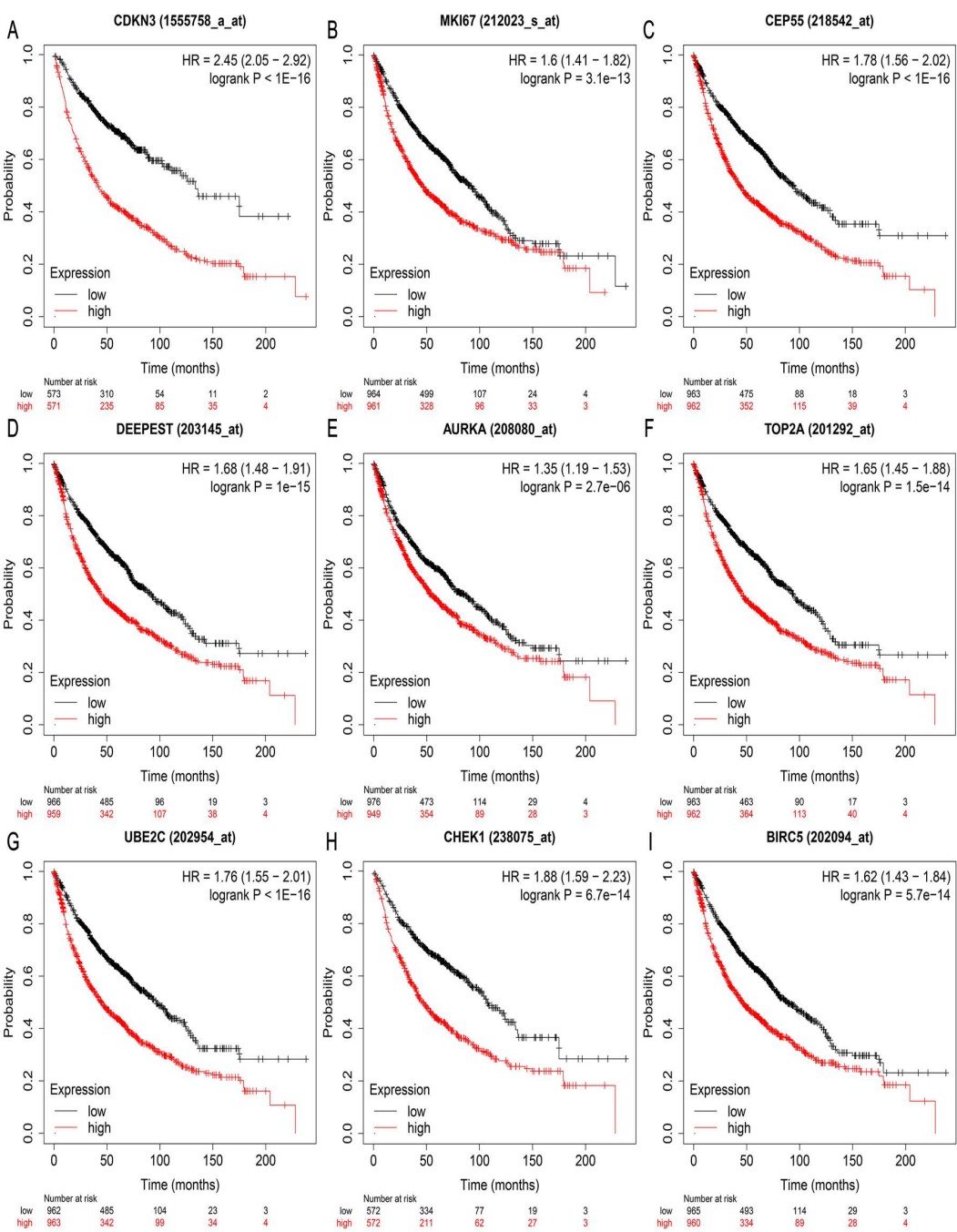

**Figure 6** **Prognostic value of (A–I)** *CDKN3, MKI67, CEP55, SPAG5 (DEEPEST), AURKA, TOP2A, UBE2C, CHEK1,* **and** *BIRC5* **in LUAD patients.** The prognostic information of the nine hub genes in patients with LUAD was from Kaplan–Meier plotter database. LUAD, lung adenocarcinoma.

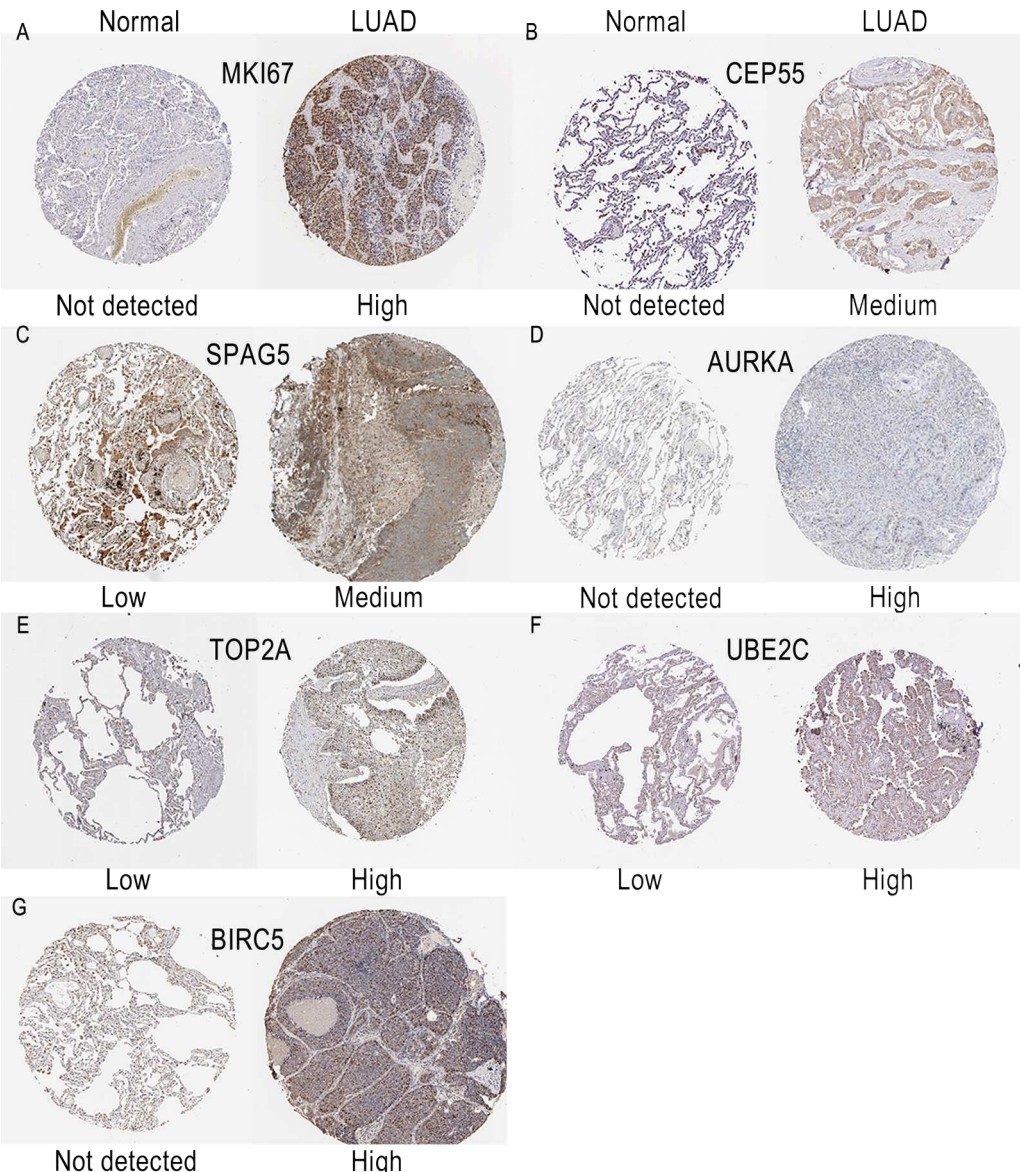

**Figure 7  Immunohistochemistry images of hub genes in LUAD tissues and normal lung tissues derived from the HPA database.** The protein images of (A–G) *MKI67, CEP55, SPAG5, AURKA, TOP2A, UBE2C* and *BIRC5* in HPA database. LUAD, lung adenocarcinoma; HPA, Human Protein Atlas.

researchers, forming a series of gene expression datasets. By integrating multiple datasets, key genes involved in the progression and prognosis of lung cancer can be fully identified (*Jin et al., 2020a*; *Jin et al., 2020b*; *Wu et al., 2020*). We analyzed GEO datasets from the Chinese lung population and used bioinformatics to discover possible biomarkers of lung cancer.

In this study, we analyzed 6 GEO datasets including GSE136043, GSE130779, GSE118370, GSE85841, GSE85716, and GSE89039, and a lot of 499 overlapping DEGs (160 upregulated and 339 downregulated genes) were identified among the datasets. The GO enrichment

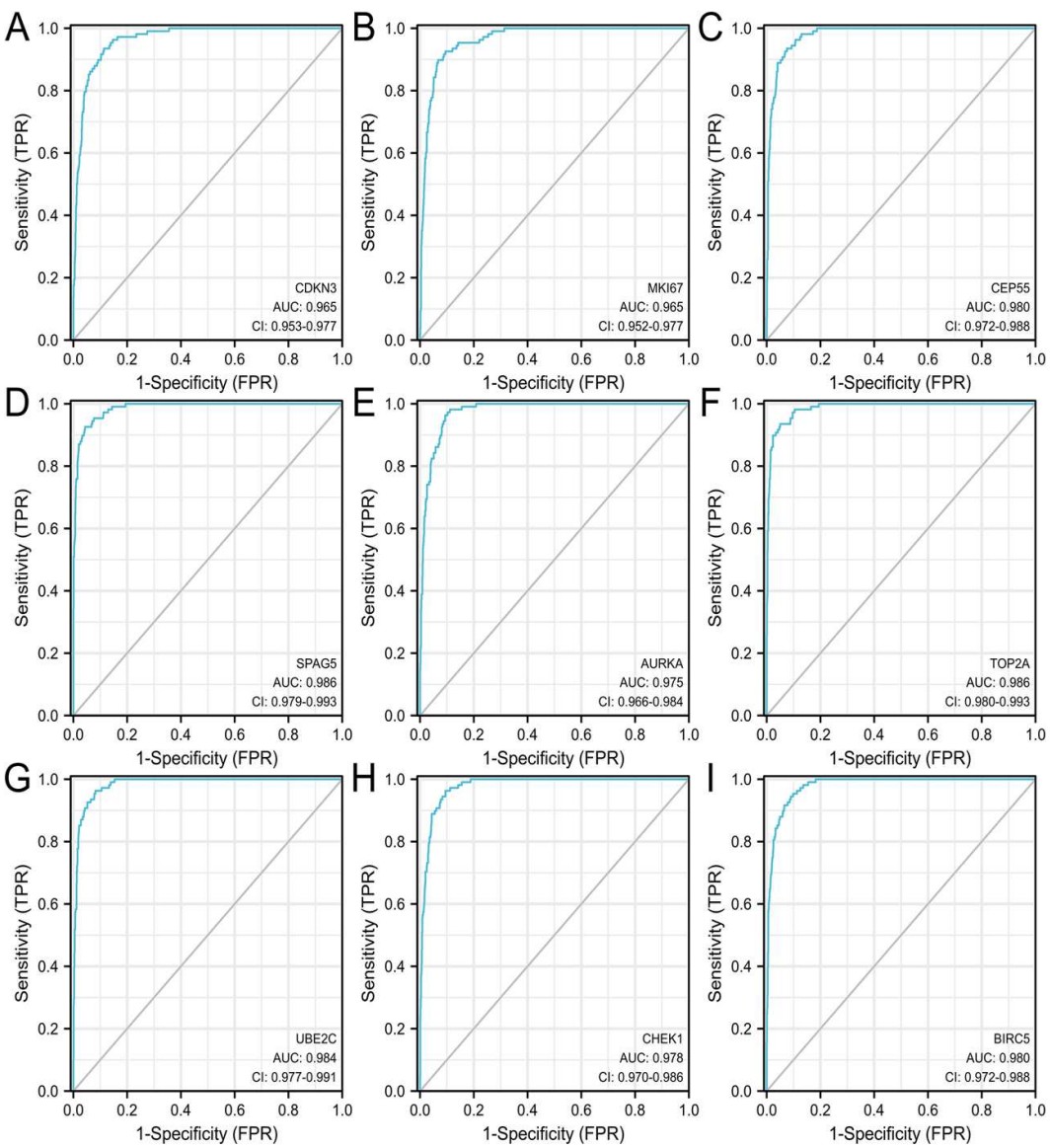

**Figure 8 ROC curves of hub genes in TCGA database.** The ROC curves of (A–I) *CDKN3, MKI67, CEP55, SPAG5, AURKA, TOP2A, UBE2C, CHEK1,* and *BIRC5*. ROC, receiver operating characteristic.

analysis indicated that the overlapping DEGs were mainly associated with angiogenesis, extracellular region, and heparin binding. Angiogenesis is a complex process, which plays a key role in maintaining tumor microenvironment, tumor growth, invasion and metastasis (*Yu & Tian, 2020*). With a large number of studies on individual proteins, heparin-binding proteins (HBPs) have been proven to be important signaling molecules in the cell microenvironment affect the basic biological processes of development, homeostasis, and diseases (*Nunes et al., 2019*). The KEGG enrichment analysis indicated that the overlapping DEGs were mainly enriched in pathways in cancer, focal adhesion, and protein digestion and absorption. We extracted nine hub genes (*CDKN3, MKI67, CEP55, SPAG5, AURKA,*

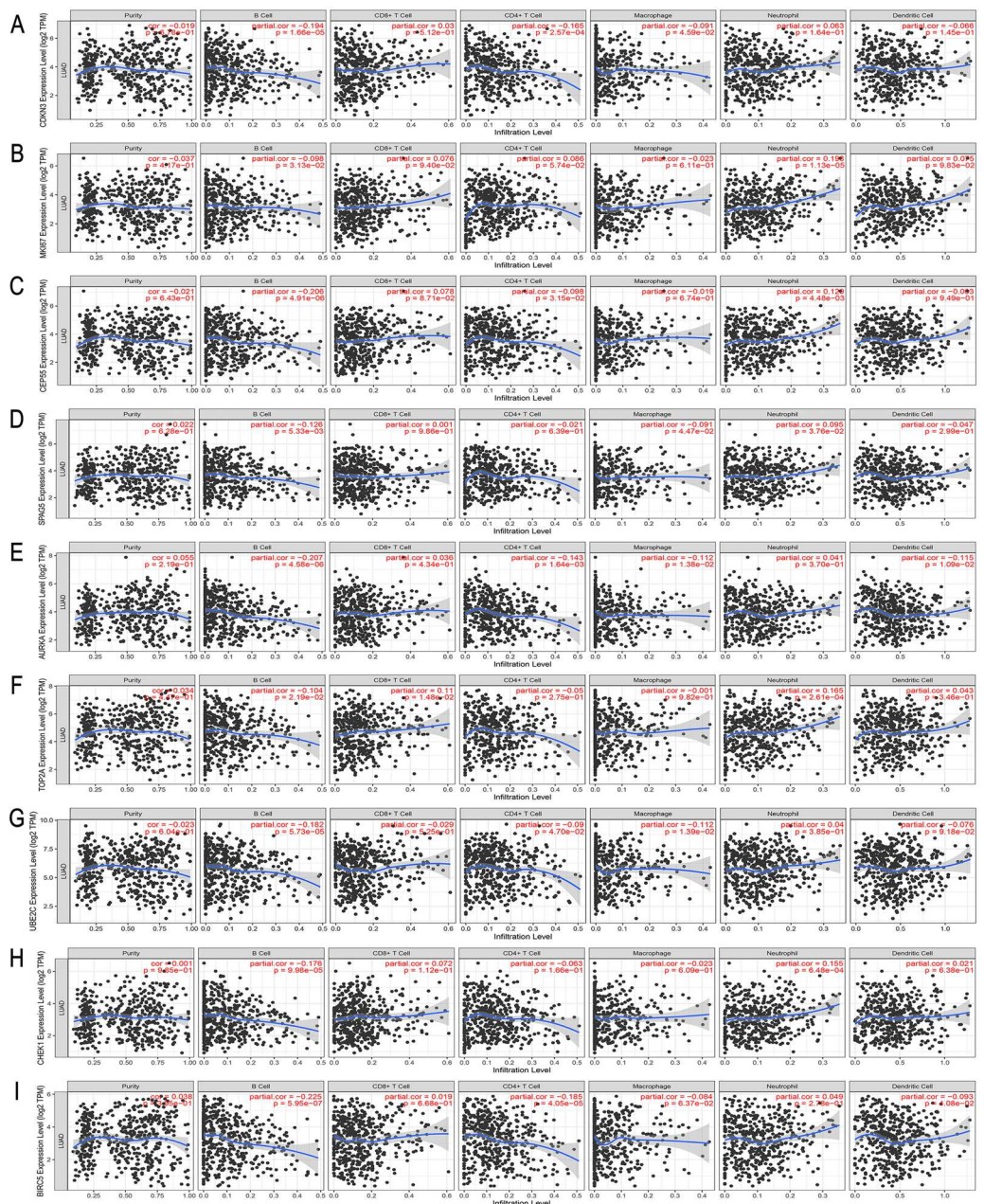

**Figure 9** Correlation between the expression of (A–I) *CDKN3, MKI67, CEP55, SPAG5, AURKA, TOP2A, UBE2C, CHEK1, BIRC5* and immune cell populations (B Cell, CD8+ T Cell, CD4+ T Cell, macrophage, neutrophil, and dendritic cell) in LUAD. $P < 0.05$ was considered statistically significant.

*TOP2A, UBE2C, CHEK1* and *BIRC5*) through PPI and module analysis. All of these genes were up-regulated in LUAD based on GEPIA database. Up-regulation of nine seven hub genes were associated with a poor prognosis of LUAD. Based on the HPA database, we found that the protein expression levels of most hub genes were higher in LUAD. Based on the ROC analysis, our results showed that all nine hub genes (*CDKN3, MKI67, CEP55,*

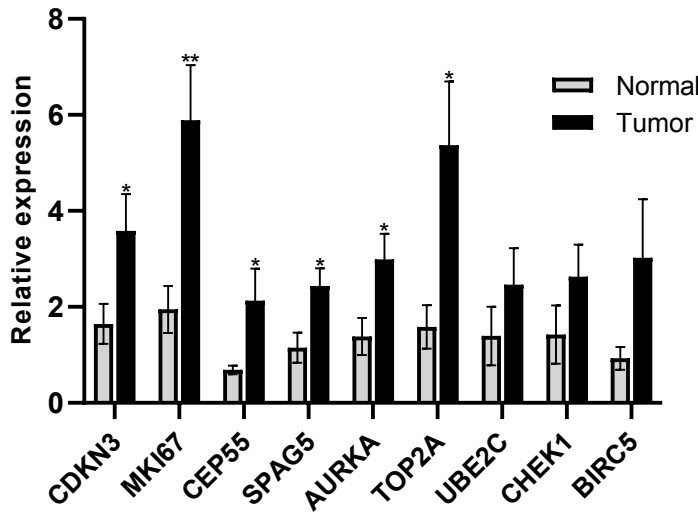

**Figure 10** RT-PCR validation of the hub genes between lung cancer tissues and normal controls (* $P <$ 0.05, ** $P < 0.01$).

*SPAG5, AURKA, TOP2A, UBE2C, CHEK1* and *BIRC5*) have good diagnostic efficiency in LUAD. The qRT-PCR analysis showed that the relative expression levels of *CDKN3, MKI67, CEP55, SPAG5, AURKA,* and TOP2A were consistent with the results of bioinformatics analysis.

The cyclin-dependent kinase inhibitor 3 (*CDKN3*) gene encodes a bi-specific protein, tyrosine phosphatase, that plays a key role in cell cycle and proliferation (*Yu et al., 2020*). *CDKN3* overexpression is prognostic of poor overall survival in lung adenocarcinoma (*Fan et al., 2015*). Ki-67 is expressed in the active phases of the cell cycle, including G1, G2 and S, and has been used as an independent biomarker to predict prognosis in patients with lung cancer (*Zheng et al., 2021*). The centromeric protein CEP55, encoded by the *CEP55*, is widely expressed in different types of tissues, especially in proliferating tissues (*Eloubeidi et al., 2002*). *CEP55* can be used as a diagnostic marker for LUAD and LUSC, but only as an independent prognostic factor for LUAD rather than LUSC (*Fu et al., 2020*). Sperm-associated antigen 5 (*SPAG5*, also known as asstrin) is involved in mitotic spindle formation and chromosome segregation, and has carcinogenic effects in tumorigenesis of various cancer types (*Huang & Li, 2020*). *AURKA* is a serine/threonine kinase that is critical for the control of mitotic progression, centrosomal maturation/separation, and mitotic spindle function (*Miralaei et al., 2021*). Studies have found that *AURKA* mRNA expression is an independent predictor of poor prognosis in patients with NSCLC (*Al-Khafaji et al., 2017*). *TOP2A*, a cycle-dependent protein, is involved in a variety of cell biological processes, such as DNA replication, chromatin condensation, chromosome separation, and chromosome structure maintenance (*Chen et al., 2015*). *TOP2A* may be a prognostic biomarker and potential therapeutic target for patients with LUAD (*Du et al., 2020*). We found that the expression levels of *UBE2C, CHEK1,* and *BIRC5* in tumor samples were

not significantly different from adjacent normal samples, which may be due to the small sample size.

Analyzing GEO datasets of different populations may find different hub genes. A GEO data analysis of the American LUAD population found six hub genes (*VIPR1, FCN3, CA4, CRTAC1, CYP4B1,* and *NEDD9*) related to prognosis (*Jiawei et al., 2020*). Another study on LUAD populations in Japan and USA found eight hub genes (*GPX3, TCN1, ASPM, PCP4, CAV2, S100P, COL1A1,* and *SPOK2*) (*Tu et al., 2021*). These genes are different from those found in our study.

There are some limitations in our research, such as small sample size, lack of experimental validation *in vivo* and *vitro*, and no consideration of clinical information. More clinical samples and molecular experiments are needed in the future to conform the function of hub genes in lung cancer.

## CONCLUSIONS

In conclusion, we filtrated a total of 499 overlapping DEGs from six GEO datasets and further validated six hub genes (*CDKN3, MKI67, CEP55, SPAG5, AURKA,* and *TOP2A*). The six hub genes were likely associated with the prognosis of lung patients in Chinese population. The functional pathways identified in the study may contribute to understand the molecular mechanisms of lung cancer. Our findings may provide new therapeutic targets for lung cancer patients.

## ACKNOWLEDGEMENTS

We thank MyGene Diagnostics Co., Ltd. for the technological assistance.

### Funding

This study was supported by the Pearl River S&T Nova Program of Guangzhou (grant no. 201906010020). The funders had no role in study design, data collection and analysis, decision to publish, or preparation of the manuscript.

### Grant Disclosures

The following grant information was disclosed by the authors:
Pearl River S&T Nova Program of Guangzhou: 201906010020.

### Competing Interests

Mengzhen Li and Zhouyu Wang are employed by MyGene Diagnostics Co., Ltd.

### Author Contributions

- Ping Liu conceived and designed the experiments, performed the experiments, analyzed the data, authored or reviewed drafts of the paper, and approved the final draft.
- Hui Li performed the experiments, analyzed the data, authored or reviewed drafts of the paper, and approved the final draft.

- Chunfeng Liao and Yuling Tang performed the experiments, prepared figures and/or tables, and approved the final draft.
- Mengzhen Li and Zhouyu Wang analyzed the data, prepared figures and/or tables, and approved the final draft.
- Qi Wu and Yun Zhou conceived and designed the experiments, authored or reviewed drafts of the paper, and approved the final draft.

## Data Availability

The data is available at Gene Expression Omnibus (GEO): GSE136043, GSE130779, GSE118370, GSE85841, GSE85716, and GSE89039.

## Supplemental Information

Supplemental information for this article can be found online at http://dx.doi.org/10.7717/peerj.12731#supplemental-information.

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
