# Peer review of "Identification of key genes and biological pathways in Chinese lung cancer population using bioinformatics analysis"

_PeerJ, doi:10.7717/peerj.12731_

## Round 0.1 · original submission · Major Revisions

The manuscript which you submitted to PeerJ has been reviewed. The reviewers have recommended publication pending major revisions which include a thorough validation of the findings. Therefore, I invite you to respond to the reviewers' comments at the bottom of this letter and revise your manuscript accordingly.

Reviewer 1 ·

Basic reporting

This article is trying to identify hub genes in Chinese lung cancer.

Experimental design

The overall design of this study is quite general.

Validity of the findings

The report of the methods needs some improvement. The validity of the results is reliable.

Additional comments

This article is trying to identify hub genes in Chinese lung cancer. The overall design of this study is quite general. The report of the methods needs some improvement. The validity of the results is reliable. The findings and conclusions were well supported.
Thus, I suggest accepting this paper. My major concerns are that the sample size is small for lung cancer research. I suggest the author add more GEO dataset which contains samples from the Chinese population
1. Line 97, please specify which R package was used.
2. Line 99, log2, or log10 fold change?

Reviewer 2 ·

Basic reporting

not good

Experimental design

not good

Validity of the findings

not good

Additional comments

Author performed reanalyzed work
GEO accession no GSE136043 was previously analysed, work done and published.
1. The roles and mechanisms of the circular RNA circ_104640 in early-stage lung adenocarcinoma: a potential diagnostic and therapeutic target (PMID: 33569440 PMCID: PMC7867959 DOI: 10.21037/atm-20-8019)
2. Combined Metabolomics with Transcriptomics Reveals Important Serum Biomarkers Correlated with Lung Cancer Proliferation through a Calcium Signaling Pathway (PMID: 34056907 DOI: 10.1021/acs.jproteome.0c01019)

GEO accession no GSE130779 was previously analysed, work done and published.
1. Identification of key genes in lung adenocarcinoma based on a competing endogenous RNA network (PMID: 33281971 PMCID: PMC7709547 DOI: 10.3892/ol.2020.12322)
2. The relationship between LncRNAs and lung adenocarcinoma as well as their ceRNA network (PMID: 33896828 DOI: 10.3233/CBM-203078)

GEO accession no GSE118370 was previously analysed, work done and published.
1. Identification of therapeutic targets and mechanisms of tumorigenesis in non-small cell lung cancer using multiple-microarray analysis (PMID: 33126319 PMCID: PMC7598833 DOI: 10.1097/MD.0000000000022815)
2. LINC00476 Suppresses the Progression of Non-Small Cell Lung Cancer by Inducing the Ubiquitination of SETDB1 (PMID: 33370431 DOI: 10.1667/RADE-20-00105.1)


GEO accession no GSE85841 was previously analysed, work done and published.
1. Prognostic value of BIRC5 in lung adenocarcinoma lacking EGFR, KRAS, and ALK mutations by integrated bioinformatics analysis (PMID: 31093306 PMCID: PMC6481100 DOI: 10.1155/2019/5451290)
2. Identification of Hub Genes Associated with Lung Adenocarcinoma Based on Bioinformatics Analysis (PMID: 31698633 DOI: 10.3934/mbe.2019384)


Re analysed work is not acceptable Meanwhile, Author not provided Differential gene expression (DEGs) table with probe id, logFC, pValue, adj.P.Val, t value and Gene Name, which are more fundamental and basic in this work. No evedance in this work without this DEGstable.
Author not performed construction and analysis of target genes - TF regulatory network and target genes - miRNA regulatory network
Author also not performed validation of hub gene ((FABP4, LPL, PDK4, and PPARGC1A ) by ROC analysis and RT-PCR.


No originality and no novelty in this work.

I strongly recommend rejection

Reviewer 3 ·

Basic reporting

There still have grammatical errors and typos in this manuscript. For examples:
- ... in progression of lung canxer ...
- ... we validated he expression levels ...
- The results of GO annotation contains three parts ...
- Some parts miss punctuation.
- ...
Therefore, the authors are suggested to re-check and revise carefully to meet the quality for publication.

Quality of figures should be improved.

Experimental design

The authors merged different data without considering batch effect removal.

Source codes should be provided for replicating the methods.

A critical concern is the use of a very small sample size in this study. This number of data cannot ensure a consistent/reliable result.

GO database or analysis has been used in previous studies i.e., https://doi.org/10.1016/j.neucom.2019.09.070 and https://doi.org/10.1016/j.csbj.2019.09.005. Therefore, the authors are suggested to refer to more works in this description.

Multivariate analysis on hub genes should be performed.

ROC analysis should be performed also.

Validity of the findings

More results/discussions on the differences between Chinese and other populations should be conducted.

There should have validation data.

The authors raised some limitations, however, they did not show how to avoid the limitations in further studies.

How to define the cut-off values when selecting the top GO terms (Table 2)?

Additional comments

No comment.

---

## Round 0.2 · Major Revisions

Please revise the manuscript as the reviewer suggested.

Reviewer 1 ·

Basic reporting

No new comments

Experimental design

No new comments

Validity of the findings

No new comments

Additional comments

No new comments

Reviewer 3 ·

Basic reporting

No comment.

Experimental design

No comment.

Validity of the findings

No comment.

Additional comments

Thanks for addressing my previous comments. However, some comments were not addressed well and there must have a room for improvement, including:
- RT-PCR should be conducted to validate the results as well as hub genes.
- My previous comment #5 were not answered.
- The meaning of immunohistochemistry images in Fig. 7 should be explained.
- What are positive and negative classes in ROC curves?
- How to define the cut-off values when selecting the top GO terms (Table 2)? Why top 5 is selected as optimal ones?

---

## Round 0.3 · accepted · Accept

This manuscript can now be accepted.

Reviewer 3 ·

Basic reporting

No comment.

Experimental design

No comment.

Validity of the findings

No comment.

Additional comments

No comment.